# The Alarmin Triad—IL-25, IL-33, and TSLP—Serum Levels and Their Clinical Implications in Chronic Spontaneous Urticaria

**DOI:** 10.3390/ijms25042026

**Published:** 2024-02-07

**Authors:** Carmen-Teodora Dobrican-Băruța, Diana Mihaela Deleanu, Ioana Adriana Muntean, Irena Nedelea, Radu-Gheorghe Bălan, Gabriela Adriana Filip, Lucia Maria Procopciuc

**Affiliations:** 1Department of Allergology and Immunology, “Iuliu Hatieganu” University of Medicine and Pharmacy, 400012 Cluj Napoca, Romania; dobrican.carmen@umfcluj.ro (C.-T.D.-B.);; 2Allergology Department, “Octavian Fodor” Institute of Gastroenterology and Hepatology, 400162 Cluj Napoca, Romania; 3Department of Anatomy, “Iuliu Hatieganu” University of Medicine and Pharmacy, 400006 Cluj-Napoca, Romania; 4Department of Biochemistry, “Iuliu Hatieganu” University of Medicine and Pharmacy, 400349 Cluj-Napoca, Romania; luciamariaprocopciuc@yahoo.com

**Keywords:** alarmins, IL-33, IL-25, TSLP, CSU, urticaria, skin, mast cells, UAS7, DLQI

## Abstract

This study delves into the critical role of alarmins in chronic spontaneous urticaria (CSU), focusing on their impact on disease severity and the quality of life (QoL) of patients. We investigated the alterations in alarmin levels in CSU patients and their correlations with the Urticaria Activity Score (UAS7) and the Dermatology Life Quality Index (DLQI). We analyzed serum levels of interleukin-25 (IL-25), interleukin-33 (IL-33), and thymic stromal lymphopoietin (TSLP) in 50 CSU patients, comparing these to 38 healthy controls. The study examined the relationship between alarmin levels and clinical outcomes, including disease severity and QoL. Elevated levels of IL-33 and TSLP in CSU patients (*p* < 0.0001) highlight their potential role in CSU pathogenesis. Although IL-25 showed higher levels in CSU patients, this did not reach statistical significance (*p* = 0.0823). Crucially, IL-33’s correlation with both UAS7 and DLQI scores underscores its potential as a biomarker for CSU diagnosis and severity assessment. Of the alarmins analyzed, IL-33 emerges as particularly significant for further exploration as a diagnostic and prognostic biomarker in CSU. Its substantial correlation with disease severity and impact on QoL makes it a compelling candidate for future research, potentially serving as a target for therapeutic interventions. Given these findings, IL-33 deserves additional investigation to confirm its role and effectiveness as a biomarker and therapeutic target in CSU.

## 1. Introduction

Chronic spontaneous urticaria (CSU), historically known also as chronic idiopathic urticaria (CIU), is a condition that has garnered increased attention in the realm of dermatological research due to its rising global prevalence. The current medical discourse, guided by the EAACI/GA2LEN/EDF/WAO guidelines, predominantly utilizes the term CSU to describe this disorder characterized by unprovoked wheals, angioedema, or both, persisting for more than six weeks. The distinctiveness of CSU lies in its extended duration and spontaneous symptom onset, setting it apart from other types of urticaria and highlighting the complexity of its pathogenesis [1,2].

The escalation in the global incidence of CSU has coincided with an evolving understanding of its etiological factors [2]. This condition intricately intertwines immunological and non-immunological elements, such as stress, pseudoallergens, autoimmunity, infection, and inflammation [3,4]. The recent COVID-19 pandemic has further spurred investigations into the potential links between viral infections, including SARS-CoV-2, and the onset or aggravation of CSU [5]. Over the years, the exploration of CSU has progressively unveiled the involvement of a broad spectrum of immune cells in its pathophysiology. Initially, significant attention was directed towards the roles of eosinophils and basophils [6,7,8] in the development and perpetuation of the disease. However, recent advancements in immunological research have shifted this focus, highlighting mast cells as the central figures in the initiation and propagation of CSU’s pathogenic mechanisms. This paradigm shift underscores the complexity of immune cell interactions in CSU, particularly emphasizing the critical role of mast cells in orchestrating the inflammatory responses [3,9,10,11]. Concurrent with the intensified scrutiny of mast cells in CSU, a broad array of cytokines associated with these cells has come under the research spotlight. This includes pro-inflammatory cytokines like IL-1 beta and IL-6, which are instrumental in driving the inflammatory cascade [12,13,14]. Additionally, IL-17, often implicated in autoimmunity, and IL-31, commonly referred to as the” pruritus cytokine”, have been identified as having significant implications in the pathogenesis and clinical manifestation of CSU [15,16]. These findings have expanded our understanding of the complex interactions between various cytokines and mast cells in the pathogenesis of CSU. Ultimately, this has led to a heightened interest in alarmins—cytokines originating from mast cells that initiate Th2 responses. Assessing the dynamics of these alarmins, including IL-25, IL-33, and TSLP, is now recognized as a critical step in understanding their role in CSU [17]. The role of mast cells extends beyond CSU, serving as essential mediators in various skin diseases through their interaction with cytokines such as alarmins [3,9,10,11]. These interactions are pivotal in the development of skin inflammation, as demonstrated by studies indicating that IL-33 can induce skin inflammation with mast cell and neutrophil activation, suggesting a broad immunological role of IL-33 in conditions like psoriasis and atopic dermatitis [18]. Similarly, the neurobeachin-like 2 protein has been identified as a regulator of mast cell homeostasis, affecting their differentiation, proliferation, and cytokine production, which is crucial for understanding mast cell functions in allergic reactions and skin diseases [19]. Furthermore, the IL-33 and IL-37 axis elucidates the balancing act between pro-inflammatory and anti-inflammatory responses in skin and allergic diseases, highlighting the therapeutic potential of targeting these pathways [20]. Moreover, the initiation of vascular responses to contact allergens by mast cells, as mediated by cell stress signals, emphasizes the significance of mast cells and alarmins in immediate and chronic inflammatory responses [21]. Despite the growing interest in alarmins, their exact roles in the pathogenesis of CSU remain underexplored. Our study aims to fill this knowledge gap by investigating the dynamics of serum alarmin levels in CSU patients and their correlations with disease severity and impact on QoL. By assessing the levels of IL-25, IL-33, and TSLP in CSU patients and comparing them with those in healthy controls, we seek to elucidate the contribution of these cytokines to the pathophysiology of CSU. Furthermore, a deep understanding of the correlation between alarmins and the clinical impact of CSU, as quantified by assessment tools like the UAS7 and the DLQI, is crucial in refining therapeutic strategies. These assessment tools are indispensable in gauging the severity and the QoL implications of CSU, thus guiding treatment approaches. This line of investigation is particularly relevant in the context of the current focus on biologic therapies and monoclonal antibodies aimed at cytokines involved in allergic and autoimmune diseases. Currently, omalizumab is the only monoclonal antibody approved for CSU, yet there exists a subset of patients who do not respond adequately to this treatment [1,2,22]. This scenario underscores the pressing need for continued research into the mechanisms of treatment resistance and factors influencing treatment response in CSU [23,24,25]. The exploration of alternative therapeutic targets and the development of novel monoclonal antibodies are pivotal in addressing these challenges and improving patient outcomes.

## 2. Results

### 2.1. Collection and Presentation of Clinical and Paraclinical Data

In this study, we meticulously gathered and analyzed essential clinical data from all participants, including demographic information and significant clinical characteristics. Additionally, we evaluated a comprehensive set of laboratory parameters and paraclinical metrics for both control subjects and CSU patients. Table 1 provides a comparative analysis of the characteristics between the two groups, presenting key demographics such as the total number of participants, gender distribution, average age (with standard deviation), and atopy status. Furthermore, the table includes critical paraclinical parameters, including IL-25, TSLP, IL-33, total IgE, Eos, RF, ESR, and CRP. This integrated approach allows for a comprehensive exploration of both clinical and laboratory aspects, shedding light on the nuanced differences between control and CSU groups.

### 2.2. Serum Alarmin Levels and ROC Analysis in CSU Patients versus Controls

Our analysis revealed a markedly significant disparity in the serum levels of alarmins (IL-25, IL-33, and TSLP) between CSU patients and control subjects.

#### 2.2.1. Analysis of IL-25, IL-33, and TSLP Serum Levels in CSU Patients and Healthy Controls

In our detailed assessment of serum alarmin levels comparing CSU patients to controls, the Mann–Whitney U test revealed distinct patterns for IL-25, IL-33, and TSLP. Specifically for IL-25, CSU patients exhibited median levels of 125.9 pg/mL against 61.94 pg/mL in controls, a difference that is not statistically significant (*p* = 0.0823) (Figure 1a). In contrast, the levels of IL-33 and TSLP in CSU patients significantly surpassed those in controls, with median levels of 160.8 pg/mL for IL-33 and 551.4 pg/mL for TSLP, sharply deviating from the control medians of 10.98 pg/mL and 396.2 pg/mL, respectively, both with *p*-values of <0.0001 (Figure 1b,c). These visual summaries present clearly different profiles for alarmin levels, providing valuable insights into their diagnostic relevance in CSU.

#### 2.2.2. Diagnostic Performance of Serum Alarmins in CSU: ROC Curve Assessment

Receiver operating characteristic (ROC) analysis was conducted to evaluate the diagnostic performance of serum IL-25 levels in differentiating CSU patients from controls. The area under the ROC curve (AUC) was calculated to be 0.6511, with a standard error of 0.06018. The 95% confidence interval for the AUC ranged from 0.5331 to 0.7690. The analysis yielded a statistically significant *p*-value of 0.0156, indicating a discriminative capacity of serum IL-25 levels between the two groups (Figure 2).

In parallel assessments, the diagnostic capacities of serum IL-33 and TSLP levels were examined through ROC curve analyses. For IL-33, the AUC was 0.9658, indicative of a high discriminative ability to distinguish CSU patients from controls, supported by a significant *p*-value of less than 0.0001 (Figure 3a). The analysis denotes IL-33 as a robust biomarker, with a narrowly defined standard error of 0.01623 and a 95% confidence interval between 0.9340 to 0.9976. Similarly, TSLP demonstrated a notable diagnostic accuracy, with an AUC of 0.7600, a standard error of 0.05048, and a 95% confidence interval from 0.6611 to 0.8589, also yielding a highly significant *p*-value of less than 0.0001 (Figure 3b). These analyses collectively underscore the potential of serum IL-33 and TSLP levels in the diagnostic stratification of CSU patients.

### 2.3. Inter-Alarmin Correlations in Serum 

Within the scope of our analysis, a detailed examination of potential correlations between the serum levels of the alarmins—IL-25, IL-33, and TSLP—was conducted. This investigation included pairwise comparisons among these cytokines to identify any interdependencies or associative patterns. However, our findings revealed that none of these inter-alarmin correlations reached statistical significance. All of the observed relationships yielded *p*-values exceeding the threshold of 0.05, indicating a lack of meaningful association. Furthermore, the correlation coefficients for these comparisons also presented as statistically non-significant, further substantiating the absence of notable interrelations among these alarmin levels in our study cohort.

### 2.4. Correlation Analysis of Serum Alarmin Levels with Inflammatory Markers in CSU

Subsequently, our investigation extended to assess correlations between each of these alarmins and various inflammatory markers, including ESR, CRP, RF, and serum eosinophil counts. Additionally, potential associations between these alarmins and clinical markers of atopy, as well as serum IgE levels in CSU patients, were explored. In all of these instances, the correlation analyses did not reveal any statistically significant relationships. The Pearson correlation coefficients obtained in these analyses were not indicative of significant correlations, affirming the absence of meaningful associations between alarmin levels and the aforementioned inflammatory and atopic markers (Figure 4).

An exception to these findings was observed in the case of TSLP. A clinically significant direct correlation between serum TSLP levels and CRP was identified, as indicated by a *p*-value of 0.0045 and a Pearson correlation coefficient (r) of 0.3954. The details of this correlation are graphically represented in Figure 5, providing a visual interpretation of the relationship between TSLP and CRP in the context of CSU.

### 2.5. Correlation between Clinical Tools (UAS7 and DLQI) in CSU

In evaluating the relationship between the urticaria activity score over 7 days (UAS7) and the dermatology life quality index (DLQI), a Pearson correlation analysis was conducted among patients with CSU. The analysis yielded a Pearson correlation coefficient (r) of 0.8176, indicating a strong positive correlation between the two clinical instruments. This suggests that as the severity of urticaria symptoms increases, as measured with the UAS7, there is a concomitant and proportional impact on the patients’ QoL, as reflected by the DLQI scores. The strength of this association is further underscored by a highly significant *p*-value of less than 0.0001. The 95% confidence interval for the correlation coefficient extends from 0.6981 to 0.8928, reinforcing the reliability of this strong correlation. Additionally, the R squared value of 0.6685 implies that approximately 66.85% of the variability in the DLQI scores can be explained by the variability in the UAS7 scores among the CSU patient cohort. These findings highlight the interdependence of symptom severity and QoL in patients with CSU, and validate the use of UAS7 and DLQI as complementary tools in clinical assessments (Figure 6).

### 2.6. Correlation between Serum Alarmin Levels and UAS7 in CSU

#### 2.6.1. Analysis of Serum IL-25 and TSLP Levels and Their Correlation with UAS7

Pearson correlation analyses for serum IL-25 and TSLP levels against UAS7 scores in CSU patients were performed. The correlation for IL-25 levels (Figure 7a) yielded an r value of 0.1629, with a 95% confidence interval from −0.1210 to 0.4221, and a *p*-value of 0.2584, indicating a minimal and non-significant correlation. Similarly, serum TSLP levels (Figure 7b) showed an r value of −0.01836 with a 95% confidence interval between −0.2952 to 0.2613, and a *p*-value of 0.8993, also suggesting no significant association. The R squared values for IL-25 and TSLP were 0.02653 and 0.0003369, respectively, underscoring the lack of predictive value for UAS7 scores in this CSU patient cohort.

#### 2.6.2. Analysis of Serum IL-33 Levels and Their Correlation with UAS7

This investigation probed for a correlation between serum IL-33 levels and UAS7 scores in CSU. The Pearson correlation coefficient (r) was determined to be 0.7510, denoting a strong positive correlation, with a highly significant *p*-value of less than 0.0001. The 95% confidence interval for this correlation extends from 0.5976 to 0.8514, indicating a high degree of precision in the relationship between these variables. Additionally, an R squared value of 0.5640 suggests that approximately 56.40% of the variability in UAS7 can be explained by the variability in serum IL-33 levels (Figure 8).

### 2.7. Correlation between Serum Alarmin Levels and DLQI in CSU

#### 2.7.1. Analysis of Serum IL-25 and TSLP Levels and Their Correlation with DLQI

In the correlation studies of serum alarmin levels and DLQI in CSU, IL-25 levels displayed a low positive but non-significant correlation with DLQI, yielding an r value of 0.2595, a *p*-value of 0.0688, and explaining minimal variance in DLQI (R^2^ = 0.06734) (Figure 9a). Concurrently, serum TSLP levels showed a very weak and negative correlation with DLQI, denoted by a Pearson r of −0.1457 and a non-significant *p*-value of 0.3128, indicating no substantial correlation with the QoL in the study population (R^2^ = 0.02122) (Figure 9b).

#### 2.7.2. Analysis of Serum IL-33 Levels and Their Correlation with DLQI

A Pearson correlation assessment revealed a significant positive relationship between serum IL-33 and DLQI scores (r = 0.7981, *p* < 0.0001), indicating a substantial association in CSU patients. The 95% confidence interval for the correlation coefficient spans from 0.6683 to 0.8808, with an R squared value of 0.6370, reflecting that over 63% of the variability in DLQI is accounted for by changes in IL-33 serum levels (Figure 10).

Given the robust correlation between serum IL-33 levels and DLQI impact, further statistical analysis was warranted. Subsequently, the Kruskal–Wallis test, a non-parametric method suitable for comparing medians across multiple groups, was employed. This test yielded a *p*-value of 0.1216, indicating no significant median differences among the groups (Figure 11a). This suggests that categorization into these specific groups does not reveal significant disparities in how serum IL-33 levels affect the DLQI scores. While the Pearson correlation indicated a significant individual-level predictor of QoL, the Kruskal–Wallis test did not reflect these differences across the grouped categories. This juxtaposition of findings highlights that IL-33 levels correlate with QoL on a continuous scale, but such associations may not be as apparent when examining group medians, possibly due to the loss of nuanced data in group categorization. 

Expanding upon the correlations identified through the Pearson correlation and the Kruskal–Wallis test, further analysis was conducted using the Mann–Whitney U test for more focused pairwise comparisons between DLQI impact categories. Although no significant difference in serum IL-33 levels was found between the moderate and important categories (*p* = 0.1876), a significant distinction was observed when comparing the moderate and very important categories (*p* = 0.0299) (Figure 11b,d). This outcome elucidates a significant disparity in IL-33 levels that aligns with the varying impact on QoL, reinforcing the association identified by the Pearson correlation. This granular approach through pairwise comparisons reveals nuances of the data, demonstrating that significant differences in IL-33 levels emerge when considering specific pairs of DLQI impact categories, even though such differences may not be apparent across a broader comparison of all categories together. Slight elevations in IL-33 serum levels were observed in the important category compared to the very important category. These differences are particularly pronounced due to the presence of outliers, as indicated in Figure 11a,c, rather than differences in group medians. This pattern reinforces the presence of significant statistical differences just between the groups at the extremes. Patients with a very important impact of the disease on their QoL exhibit significantly higher serum IL-33 levels than those with a moderate impact, as clearly demonstrated in Figure 11d. These findings further substantiate the role of IL-33 levels as a marker reflective of the disease’s impact severity on patient QoL.

## 3. Discussion

The pivotal role of mast cells in the pathogenesis CSU is well-established and underscored by numerous recent studies [3,9,10,11,15,17]. This recognition justifies and heightens the scientific interest in focusing our attention on the cytokines that activate these cells, namely alarmins (IL-25, IL-33, and TSLP). The involvement of such cytokines in mast cell activation and their potential to exacerbate CSU has been increasingly illuminated by research [3,9,10,11,15,17]. This divergence emphasizes the critical contribution of alarmins to the mechanism of the disease. 

The potential for targeting these cytokines to mitigate the symptoms of CSU represents an intriguing and scientifically promising avenue for exploration. While the effectiveness of such therapeutic interventions has yet to be definitively proven, the development of biologic therapies aimed at these pathways, including tezepelumab (anti-TSLP), etokimab, and itepekimab (both targeting IL-33), as well as astegolimab and GSK3772847 (targeting the ST2 subunit of the IL-33 receptor), are currently undergoing clinical evaluation [26,27]. This innovative therapeutic strategy marks a significant shift in the approach to managing CSU, offering the potential to transform patient outcomes by addressing the underlying mechanisms of the disease directly. The exploration of these biologics in the clinical setting is poised to advance our understanding and treatment of CSU, reflecting a novel paradigm in our therapeutic arsenal against this challenging condition. 

This study examined the serum levels of alarmins to elucidate their association with the clinical severity of CSU and its impact on patient QoL. Our analysis indicates that IL-33 levels are notably higher in CSU patients, correlating with increased disease severity and diminished QoL as reflected by the UAS7 and DLQI scores. These observations suggest that IL-33’s influence in CSU extends beyond that of a biomarker, acting as a pivotal mediator in the disease’s pathogenesis and affecting both clinical outcomes and patient well-being. The significant correlation between IL-33 with the UAS7 and DLQI scores underlines the necessity for developing IL-33-targeted therapeutic approaches, especially for patient subgroups inadequately managed by current treatments, such as omalizumab [26,27]. The findings by Manti et al. [28] support this approach, emphasizing the potential of monoclonal antibodies in enhancing CSU management.

Consistent with the insights provided by Puxeddu et al. [29] and further elaborated in reviews by Cayrol and Girard [30], our results confirm IL-33’s essential role in the immunological cascade of CSU, highlighting its extensive biological relevance across various immunological conditions. Trier et al. [31] have identified IL-33’s capacity to aggravate histaminergic itch, a prevalent CSU symptom, suggesting a potential interaction with sensory pathways that exacerbates itch severity. This underscores IL-33’s dual functionality in both immunological responses and neurosensory processes, expanding its potential as a therapeutic target in CSU and similar pruritic conditions [32].

Furthermore, the review by Murdaca et al. [33] on the IL-33/IL-31 axis sheds light on the multifaceted nature of IL-33, proposing its dual role as a novel biomarker and therapeutic target for Th2-driven diseases. These perspectives are particularly relevant considering our data, which corroborates IL-33’s critical involvement in CSU. By integrating these findings, our research not only supports but also expands upon the existing literature, reinforcing the argument for IL-33-targeted therapies as a promising avenue to transform CSU management and patient outcomes.

In our investigation, IL-25’s elevated presence in the serum of CSU patients, while notable, did not translate into a significant statistical impact, presenting an intriguing facet in the complex tapestry of CSU pathogenesis. This contrast in our findings could indicate a nuanced role for IL-25 in CSU, potentially contributing to the condition in a less direct or potent manner than the pronounced effects seen with IL-33. The interplay of IL-25 in the context of CSU may involve intricate immune interactions that attenuate its measurable impact on disease severity and patient-reported QoL.

The nuanced role of IL-25 within the immune system reflects the complex intercellular signaling regulated by alarmin cytokines, pivotal for skin homeostasis and implicated in allergic inflammation processes. The analysis by Hasegawa et al. [34] positions IL-25 as crucial in both promoting allergic skin inflammation and facilitating wound healing, underscoring its intricate function in skin disorders. Additionally, Stanbery et al. [35] expand the scope of IL-25’s impact, illustrating its involvement in a broader spectrum of immune responses, including those against viral infections and in cancer, thus highlighting IL-25’s versatility as a cytokine affecting diverse immune mechanisms.

This broader understanding suggests that the serum IL-25 levels observed in our CSU study, while not showing a direct statistical correlation, contribute to a more comprehensive narrative. Documented increased IL-25 expression in lesional skin by Kay et al. [17], in conjunction with the regulatory roles of type 2 alarmin cytokines in skin immunity, as outlined by Hasegawa et al. [34] and the varied functions of IL-25 in tissue immunity highlighted by Stanbery et al. [35], suggest IL-25’s nuanced involvement in CSU. This prompts further investigation into IL-25’s specific contributions to CSU pathophysiology and symptomatology.

Moreover, our research adds to the ongoing dialogue on the identification of effective biomarkers for CSU, with IL-33 demonstrating significant potential as indicated by ROC curve analysis. Conversely, IL-25’s weaker performance, despite its statistical relevance, hints at a more limited role when considered in isolation, emphasizing the need for a multifaceted approach in biomarker development and the utility of these cytokines in clinical assessments.

In our study, we observed elevated levels of TSLP in CSU, aligning with findings from other studies such as those reported by Hoy et al. [36]. This observation underscored TSLP’s role in the inflammation associated with CSU, although its direct correlation with disease severity and patient QoL, as measured by DLQI and UAS7 indices, remained minimal, as discussed by Wang and Zuo [37]. The association between TSLP and CRP highlighted its involvement in generalized inflammatory responses, without a clear link to specific CSU severity or QoL outcomes.

The development and approval of tezepelumab, an anti-TSLP monoclonal antibody, underscored by Damask et al. [38], marked a pivotal advancement in targeting this cytokine for CSU treatment, offering new hope for patients who are unresponsive to standard treatments. Further research by Hashimoto et al. [39] suggested TSLP’s involvement in CSU’s pruritic symptoms through basophil activation, extending its impact to sensory mechanisms within the disease. Observations by Treudler and Simon [40] on emerging biologics targeting TSLP indicated potential for broader therapeutic applications in addressing allergic and immunological facets of CSU.

However, contrasting evidence by Metz et al. [41] regarding TSLP levels in CSU necessitated additional research to resolve discrepancies in cytokine profiles and their implications for clinical manifestations and QoL in CSU patients. Collectively, these insights [36,37,38,39,40,41] called for an integrated approach to CSU therapy that considered TSLP’s multifaceted role in the immune response, advocating for refined strategies in patient care and management.

The absence of inter-alarmin correlations suggests independent pathways of action for these cytokines in CSU pathogenesis. This is particularly notable, as it challenges the notion of a synergistic alarmin network and prompts further inquiry into these cytokines’ discrete roles. Alarmin cytokines, such as IL-25, IL-33, and TSLP, are secreted by various immune and epithelial cells in response to tissue damage, infection, or inflammation. While they are known to play critical roles in immune responses, the exact order and mechanisms of their secretion within the immune cascade remain a subject of ongoing investigation. 

The correlation between clinical tools—UAS7 and DLQI—demonstrates a robust link between symptom severity and QoL, affirming the interdependence of these measures in CSU. This strong correlation is especially significant, as it validates the clinical relevance of these tools in both research and practice.

While CSU and atopic dermatitis (AD) can be clinically distinguished by specialist clinicians, they share significant similarities in their inflammatory mechanisms, particularly in the role of pro-inflammatory cytokines like IL-4, IL-5, IL-13, and IL-31. This intersection underscores the necessity of mentioning this second condition here, allowing for a comparative and parallel overview to enhance our understanding of their shared pathophysiological features in an academic context. These cytokines, implicated in the onset of pruritus—a common symptom in both conditions—underscore the potential for a unified approach to therapy. Biologics such as dupilumab, targeting the IL-4 receptor to block type-2 inflammation, mepolizumab against IL-5, and tezepelumab and etokimab targeting TSLP and IL-33, respectively, alongside nemolizumab, which focuses on the IL-31 receptor crucial for mediating itch, represent pivotal advancements. These developments offer targeted relief from the inflammatory and pruritic aspects of AD and CSU [42]. 

By engaging with these existing contributions to the field, our study not only corroborates the established narratives, but also expands upon them, suggesting new avenues for research and potential therapeutic interventions that leverage the systemic nature of alarmin activity. It is within this expanded, complex framework that our study positions itself, aiming to bridge the gaps in current knowledge and inspire a forward momentum in both research and clinical practice.

### Strengths and Limitations

This study’s exploration into the serum levels of IL-25, IL-33, and TSLP offers valuable insights into CSU pathophysiology. One of the main strengths is the detailed examination of the correlation between these cytokines and both CSU severity and QoL outcomes, which enhances our understanding of the disease’s underlying mechanisms. The inclusion of a control group adds robustness to our findings, and the rigorous statistical analyses employed substantiate the strength of the observed associations.

However, there are limitations to consider. The cross-sectional nature of the study precludes the establishment of causality, and does not capture the longitudinal dynamics of cytokine levels. While IL-33’s elevation in CSU patients suggests a significant role, the absence of a corresponding increase in TSLP levels in relation to QoL measures points to the complex interplay of cytokines in CSU.

The study’s sample size, though sufficient for our analysis, may not reflect the full spectrum of the CSU population, and the single-center design limits the generalizability of the results. This highlights the necessity for larger, multi-center studies to confirm these findings across a more diverse patient cohort.

In conclusion, our findings provide a solid foundation for the potential targeting of IL-33 in CSU management, given its strong correlation with disease severity and QoL. Yet, the intricate web of cytokine interactions and their impact on CSU remains an area ripe for further investigation. Future research, ideally involving multi-center longitudinal studies with larger participant pools, is essential to advance our comprehension of cytokine activity in CSU and to guide therapeutic interventions.

## 4. Materials and Methods

This retrospective, analytical study was carried out at the Allergology Department of the Regional Institute of Gastroenterology and Hepatology, Cluj-Napoca, Romania. It involved a cohort of 50 CSU patients, diagnosed according to the latest international guidelines [1]. These guidelines describe CSU as a recurrent, maculopapular rash that may include angioedema, appearing at least bi-weekly for over six weeks [1,2]. For comparative analysis, 38 healthy staff members from the institute were selected as a control group. The CSU patients met diagnostic criteria as per the guidelines [1], and were free from concurrent systemic illnesses such as systemic mastocytosis, Schnitzler syndrome, and urticarial vasculitis. Additional exclusion criteria included patients with renal, hepatic, psychiatric, or infectious conditions presenting with cutaneous manifestations or itching. The control group, with same distribution by age and gender, was strictly composed of individuals with no history of urticaria and excluded those with systemic diseases causing urticaria or pruritus. This study received ethical approval from the “Iuliu Hatieganu” University of Medicine and Pharmacy, Cluj-Napoca, Romania (AVZ270/10.10.2022), and the IRGH (12637/11.10.2022). Informed consent was obtained from all participants. Demographic information and baseline characteristics such as age, sex, and serum levels of the alarmins IL-33, IL-25, and TSLP were recorded for all participants. For CSU patients, additional data including disease duration, severity, and presence or absence of atopy were collected. This was defined by a positive skin-prick (SPT) test to environmental allergens. Complementary tests included complete blood counts, ERS and CRP analysis, coproparasitological exams, and total serum IgE measurements, all conducted at the hospital’s central laboratory. Venous blood was collected from the participants and centrifuged, and the serum was stored at −80 °C. The levels of IL-33, IL-25, and TSLP in the serum were determined using specific ELISA kits, following the manufacturer’s instructions (Elabscience, 14780 Memorial Drive, Suite 108, Houston, TX, 77079, USA). 

The Urticaria Activity Score over 7 days (UAS7) is a comprehensive, globally recognized metric for evaluating the severity of CSU [43,44]. This tool captures the extent and impact of urticarial symptoms over a week, quantifying both hive formation and itch intensity. The UAS7 incorporates two distinct subscores, one for hives and the other for itch severity, each evaluated daily. Hive Score: This is assessed on a scale from 0 to 3, where 0 denotes the absence of hives, 1 signifies mild hives (less than 20 hives in a 24-hour period), 2 indicates moderate hives (between 20 and 50 hives), and 3 reflects severe hives (over 50 hives or extensive confluent areas). Itch Severity Score: Similar to the hive score, itch severity is measured on a scale from 0 to 3. A score of 0 indicates no itching, 1 represents mild itching (noticeable but not bothersome), 2 is for moderate itching (disturbing but not interfering significantly with daily activities or sleep), and 3 signifies severe itching (severely disruptive to daily activities and sleep). The total UAS7 score is the sum of these daily assessments, with a maximum potential score of 42. The cumulative score categorizes CSU severity into three levels: mild (0–15), moderate (16–27), and severe (28–42), enabling a nuanced understanding of the disease’s impact on the patient.

The Dermatology Life Quality Index (DLQI) is a widely employed instrument to gauge the impact of CSU on a patient’s QoL [45,46]. This index consists of ten questions, addressing various aspects of daily life that may be affected by the condition. Each question is scored from 0 to 3, based on the degree of impact. The questions cover a broad spectrum of life domains, including symptoms and feelings, daily activities, leisure, work or school performance, personal relationships, and treatment-related issues. The cumulative score from these questions provides a total DLQI score, ranging from 0 (no impact) to 30 (extremely large impact). This total score is further broken down into five categories: 0–1 (no impact), 2–5 (small impact), 6–10 (moderate impact), 11–20 (very large impact), and 21–30 (extremely large impact). Thus, the DLQI offers an insightful, patient-centered perspective on how CSU affects various facets of an individual’s life, complementing the clinical severity assessed using the UAS7.

### Statistical Analysis

Our data are presented predominantly as medians and interquartile ranges. We employed the Shapiro–Wilk test to evaluate the normal distribution of the data. For group comparisons, the study utilized the Mann–Whitney U test or the Kruskal–Wallis test, based on the data characteristics. Specifically, the Mann–Whitney U test was applied for comparing two distinct groups, such as the serum levels of IL-25, IL-33, and TSLP between healthy controls and CSU patients. This test is particularly suited for non-normally distributed continuous data and facilitates the assessment of significant differences in the groups’ central tendencies.

For comparisons involving more than two independent groups or ordinal data, the Kruskal–Wallis test was the method of choice. This test extends the Mann–Whitney U test to multiple groups, and assesses if there are significant differences in their medians. In our research, it was particularly useful for analyzing IL-25, IL-33, and TSLP levels across various disease severity groups.

Categorical data comparisons were conducted using Pearson’s chi-squared test, suitable for examining associations between categorical variables. Correlations between various parameters were examined using Spearman’s correlation, a non-parametric measure ideal for assessing non-linear relationships.

Furthermore, we explored the diagnostic potential of IL-25, IL-33, and TSLP using receiver operating characteristic (ROC) curve analysis. The area under the curve (AUC) for these cytokines was calculated to determine their effectiveness in distinguishing CSU patients from healthy controls.

All statistical analyses and graph generation were conducted using GraphPad Prism 9.0 software (GraphPad Software Inc., San Diego, CA, USA). A *p*-value of <0.05 was set as the threshold for statistical significance. This comprehensive selection of tests and methods was aimed at thoroughly analyzing the interrelationships and disparities in IL-25, IL-33, and TSLP levels and their clinical correlates within the context of CSU.

## 5. Conclusions

Our research into the alarmin cytokines IL-25, IL-33, and TSLP has identified a notable elevation of IL-33 in CSU patients, correlating significantly with both UAS7 and DLQI scores, thereby substantiating its role in disease severity and patient QoL impact. In contrast, the elevated serum levels of IL-25 did not correspond with significant clinical correlations, suggesting a more intricate role within the cytokine interplay of CSU. Elevated TSLP levels, while indicative of its involvement in allergic and immunological pathways, did not show a direct correlation with symptom severity or patient QoL in CSU. The study acknowledges the recent approval of tezepelumab, an anti-TSLP therapy, for other allergic and immunological disorders, which may inform future therapeutic strategies in CSU management. The discrepancies in cytokine profiles, particularly concerning TSLP, underscore the imperative for continued research to elucidate their roles in CSU. Overall, these findings advocate for further exploration into cytokine-modulating treatments, with IL-33 presenting a viable target. Future research, ideally through larger, multi-center longitudinal studies, is essential to deepen our understanding of cytokine dynamics in CSU and optimize therapeutic interventions.

## Figures and Tables

**Figure 1 ijms-25-02026-f001:**
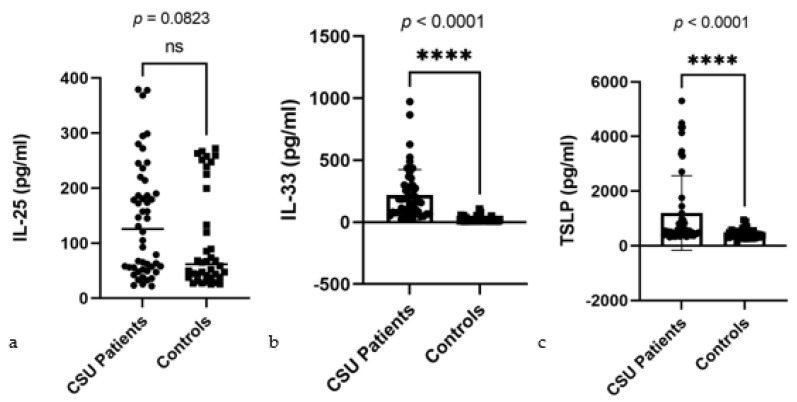
Serum alarmin levels in CSU patients and controls: (**a**) IL-25—interleukin 25, CSU—chronic spontaneous urticaria, ns—non-significant, *p* = 0.0823; (**b**) IL-33—interleukin 33, CSU—chronic spontaneous urticaria, **** = *p* < 0.0001; (**c**) TSLP—thymic stromal lymphopoietin, CSU—chronic spontaneous urticaria, **** = *p* < 0.0001.

**Figure 2 ijms-25-02026-f002:**
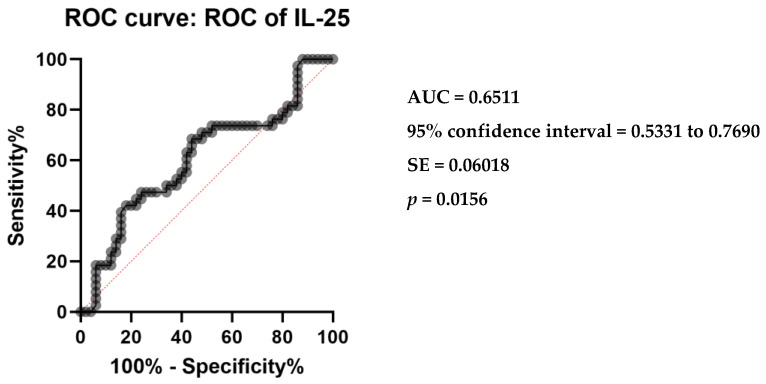
Receiver operating characteristics (ROC) curve of IL-25 in CSU compared with healthy controls; IL-25—interleukin 25, CSU—chronic spontaneous urticaria, *p* = 0.0156, AUC—area under the curve = 0.6511, and SE—standard error = 0.06018.

**Figure 3 ijms-25-02026-f003:**
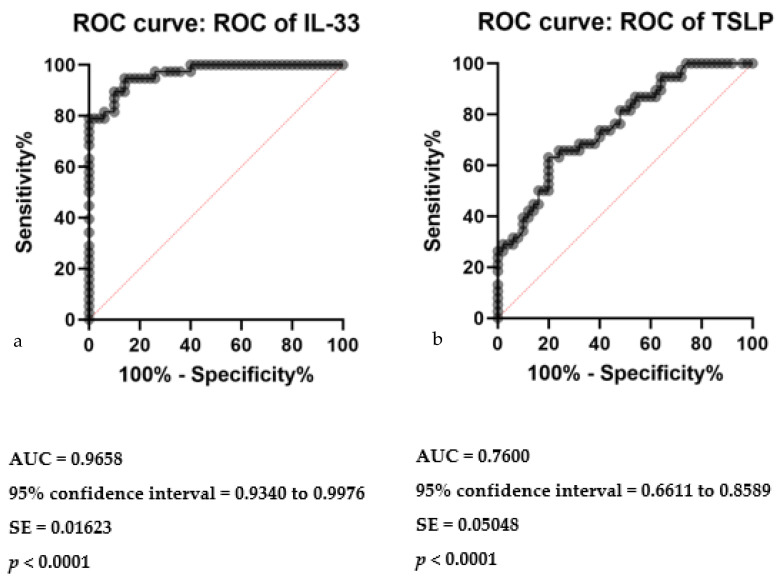
ROC curve analysis of serum IL-33 and TSLP in CSU: (**a**) Receiver operating characteristics (ROC) curve of IL-33 in CSU compared with healthy controls; IL-33—interleukin 33, CSU—chronic spontaneous urticaria, *p* < 0.0001, AUC—area under the curve = 0.9658, SE—standard error = 0.01623. (**b**) Receiver operating characteristics (ROC) curve of TSLP in CSU compared with healthy controls; TSLP—thymic stromal lymphopoietin, CSU—chronic spontaneous urticaria, *p* < 0.0001, AUC—area under the curve = 0.7600, SE—standard error = 0.05048.

**Figure 4 ijms-25-02026-f004:**
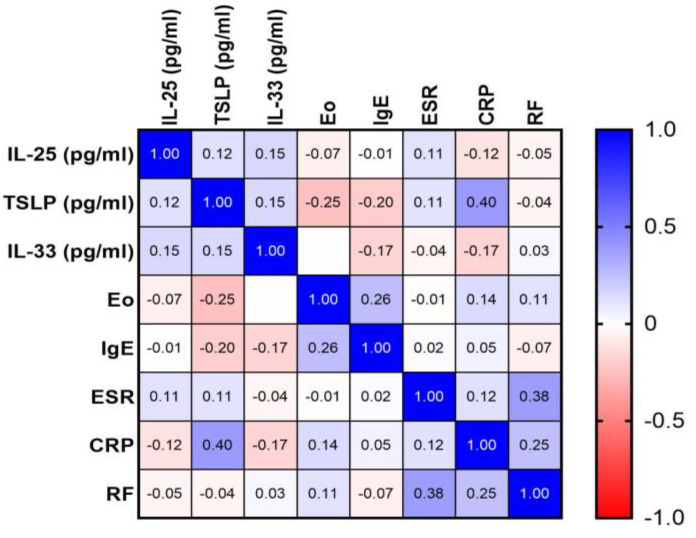
Correlation matrix of alarmin and inflammatory markers in CSU; CSU—chronic spontaneous urticaria, IL-25—interleukin-25, TSLP—thymic stromal lymphopoietin, IL-33—interleukin-33, Eo—eosinophil count, IgE—immunoglobulin E, ESR—erythrocyte sedimentation rate, CRP—C-reactive protein, RF—rheumatoid factor.

**Figure 5 ijms-25-02026-f005:**
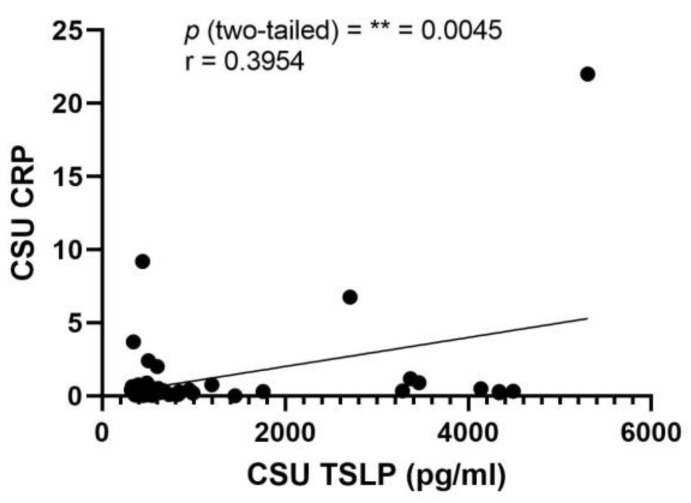
Correlation of serum TSLP and CRP in CSU; TSLP—thymic stromal lymphopoietin, CRP—C-reactive protein, CSU—chronic spontaneous urticaria, *p* = ** = 0.0045; r—the Pearson correlation coefficient for the serum level of TSLP and CRP is 0.3954 (r = 0.3954).

**Figure 6 ijms-25-02026-f006:**
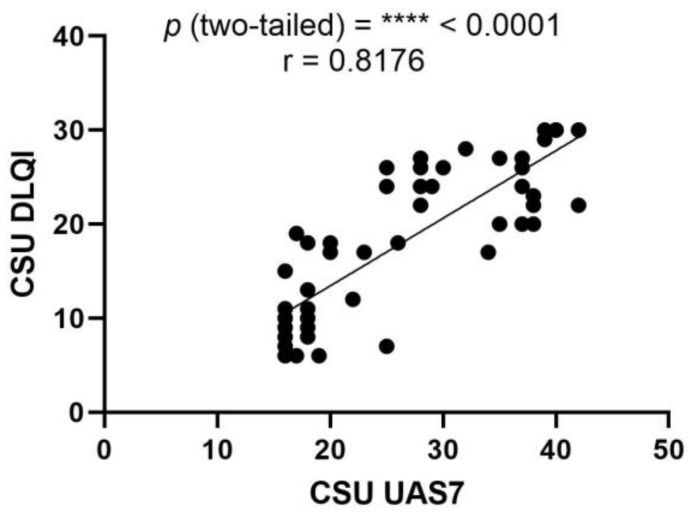
Correlation between UAS7 and DLQI in CSU; UAS7—Urticaria Activity Score over 7 days, DLQI—Dermatology Life Quality Index, CSU—chronic spontaneous urticaria, *p* = **** < 0.0001; r—the Pearson correlation coefficient for the UAS7 and DLQI is 0.8176 (r = 0.8176).

**Figure 7 ijms-25-02026-f007:**
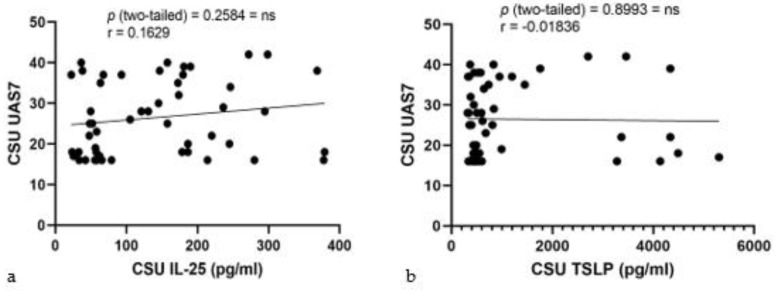
Correlation of serum IL-25 and TSLP levels with UAS7 in CSU: (**a**) Serum IL-25 levels and UAS7 in CSU; IL-25—interleukin-25, UAS7—Urticaria Activity Score over 7 days, CSU—chronic spontaneous urticaria, *p* = ns (not significant) = 0.2584; r—the Pearson correlation coefficient for serum IL-25 levels and UAS7 scores is 0.1629 (r = 0.1629). (**b**) Serum TSLP levels and UAS7 in CSU; TSLP—thymic stromal lymphopoietin, UAS7—Urticaria Activity Score over 7 days, CSU—chronic spontaneous urticaria, *p* = ns (not significant) = 0.8993; r—the Pearson correlation coefficient for serum TSLP levels and UAS7 scores is −0.01836 (r = −0.01836).

**Figure 8 ijms-25-02026-f008:**
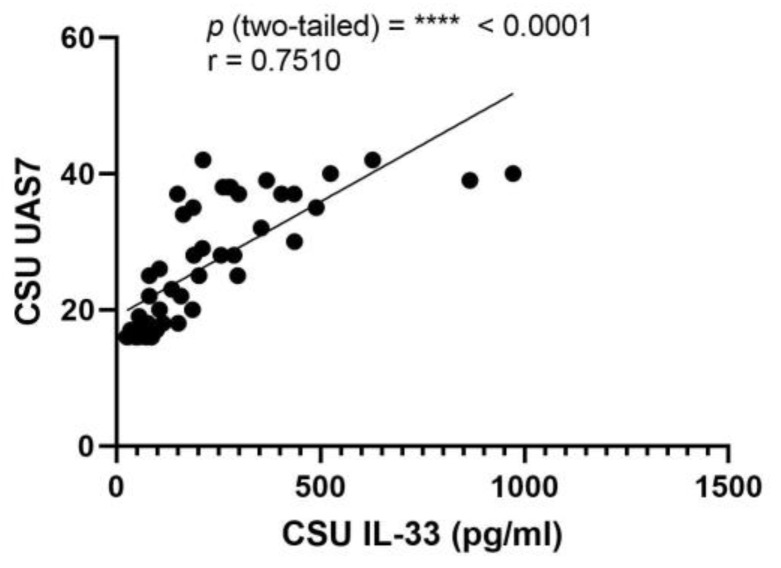
Correlation between serum IL-33 levels and UAS7 in CSU; IL-33—interleukin-33, UAS7—Urticaria Activity Score over 7 days, CSU—chronic spontaneous urticaria, *p* = **** < 0.0001; r—the Pearson correlation coefficient for serum IL-33 levels and UAS7 scores is 0.7510 (r = 0.7510).

**Figure 9 ijms-25-02026-f009:**
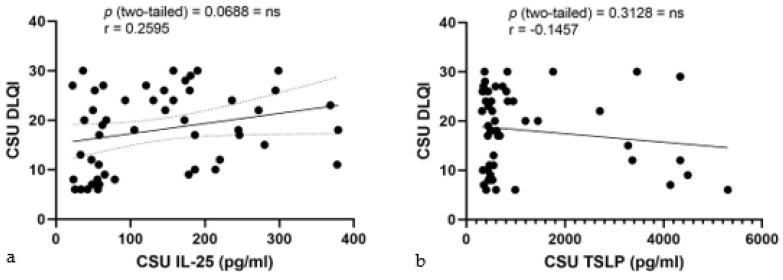
Correlation of serum IL-25 and TSLP levels with DLQI in CSU: (**a**) Correlation between serum IL-25 levels and DLQI in CSU; IL-25—interleukin-25, DLQI—Dermatology Life Quality Index, CSU—chronic spontaneous urticaria, *p* = ns (not significant) = 0.0688; r—the Pearson correlation coefficient for serum IL-25 levels and DLQI scores is 0.2595 (r = 0.2595). (**b**) Correlation between serum TSLP levels and DLQI in CSU; TSLP—thymic stromal lymphopoietin, DLQI—Dermatology Life Quality Index, CSU—chronic spontaneous urticaria, *p* = ns (not significant) = 0.3128; r—the Pearson correlation coefficient for serum TSLP levels and DLQI scores is −0.1457 (r = −0.1457).

**Figure 10 ijms-25-02026-f010:**
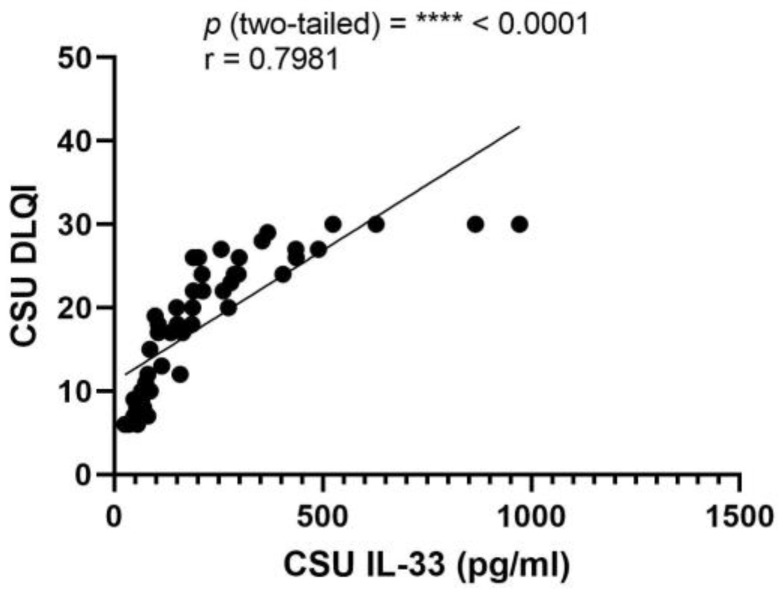
Correlation between serum IL-33 Levels and DLQI in CSU; IL-33—interleukin-33, DLQI—Dermatology Life Quality Index, CSU—chronic spontaneous urticaria, *p =* **** < 0.0001; r—the Pearson correlation coefficient for serum IL-33 levels and DLQI scores is 0.7981 (r = 0.7981).

**Figure 11 ijms-25-02026-f011:**
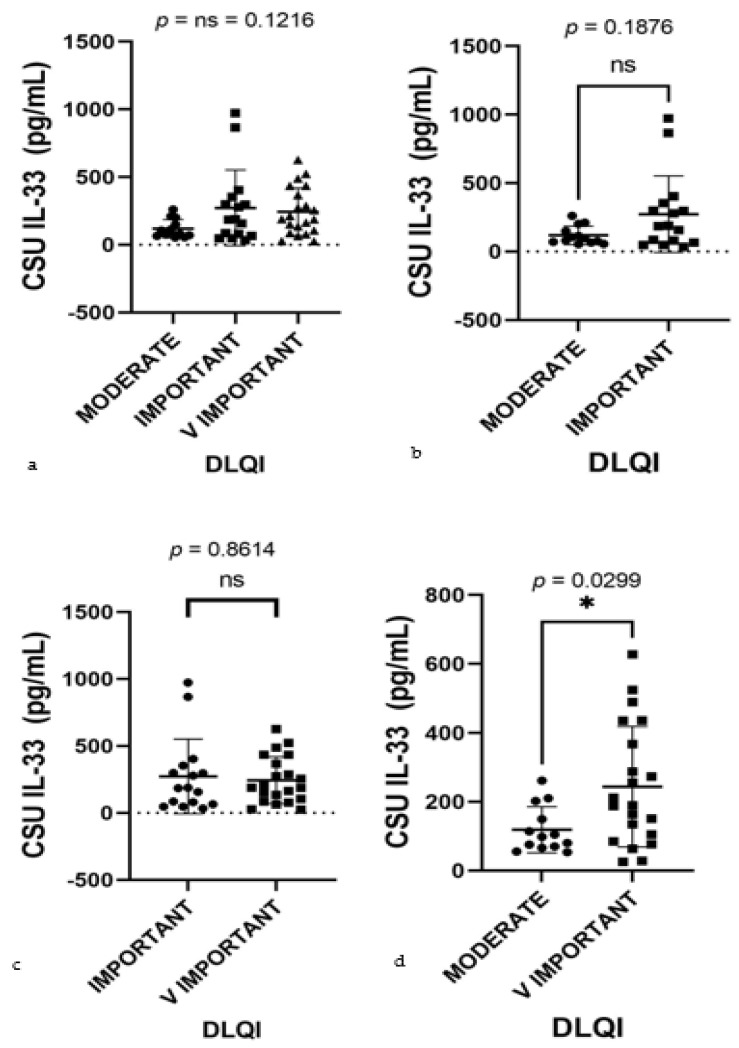
Serum IL-33 levels by DLQI classes in CSU patients; IL-33—interleukin-33, DLQI—Dermatology Life Quality Index (the DLQI classes are demarcated as moderate, important, and very important, illustrating the gradation of urticaria’s impact on patients’ lives), CSU—chronic spontaneous urticaria. (**a**) Moderate, important, and very important DLQI classes, *p* = ns (not significant) = 0.1216. (**b**) Moderate and important DLQI classes, showing no significant difference in serum IL-33 levels, with a *p* value of ns (not significant) = 0.1876. (**c**) Important and very important DLQI classes, *p* = ns (not significant) = 0.8614. (**d**) Moderate and very important DLQI classes, revealing a significant difference in serum IL-33 levels, with a *p* value of 0.0299 (* indicates statistical significance).

**Table 1 ijms-25-02026-t001:** Overview of participant characteristics.

Characteristics	Controls	CSU Patients	*p*-Values (Mann–Whitney U)
Total Participants (n)	33	50	N/A
Average Age (yrs. ± SD)	44.32 ± 9.23	50.14 ± 16.10	N/A
Gender Distribution (F/M)	26/12	36/14	N/A
Atopy Status (Atopic/Non-atopic)	10/28	14/36	N/A
IL-25 (pg/mL ± SD)	105.03 ± 89.21	140.27 ± 100.16	0.0823
TSLP (pg/mL ± SD)	434.57 ± 169.43	1200.42 ± 1348.36	<0.0001
IL-33 (pg/mL ± SD)	21.70 ± 22.68	220.67 ± 201.17	<0.0001
Total IgE (IU/L ± SD)	41.23 ± 27.08	168.63 ± 178.99	<0.0001
Eos (×1000 cells/µL ± SD)	0.163 ± 0.102	0.495 ± 0.704	0.1015
RF (IU/mL ± SD)	10.75 ± 2.82	14.02 ± 1.58	<0.0001
ESR (mm/h ± SD)	12 ± 6.71	14.29 ± 9.89	0.8074
CRP (mg/dL ± SD)	0.36 ± 0.33	0.36 ± 0.27	0.7408

Note: CSU, chronic spontaneous urticaria; n, number; yrs, years; F, female; M, male; IL-25, interleukin-25; TSLP, thymic stromal lymphopoietin; IL-33, interleukin-33; pg/mL, picograms per milliliter; total IgE, total immunoglobulin E; IU/L, international units per liter; Eos, absolute eosinophil count; cells/µL, cells per microliter of blood; RF, rheumatoid factor; IU/mL, international units per milliliter; ESR, erythrocyte sedimentation rate; mm/h, millimeters per hour; CRP, C-reactive protein; mg/dL, milligrams per deciliter. Age, IL-25, TSLP, IL-33, total IgE, Eos, RF, ESR, and CRP are presented as mean ± standard deviation (SD). *p*-values indicate the statistical significance of differences between CSU patients and controls, with *p* < 0.05 considered significant. N/A, not applicable, is used where statistical comparison was not relevant or was not conducted, such as for total participant count, average age, atopy status, and gender distribution.

## Data Availability

All data generated or analyzed during this study are included in this article. Further enquiries can be directed to the corresponding author.

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
