# Peer review of "The Alarmin Triad—IL-25, IL-33, and TSLP—Serum Levels and Their Clinical Implications in Chronic Spontaneous Urticaria"

_ijms, 2024, doi:10.3390/ijms25042026_

Round 1

Reviewer 1 Report

Comments and Suggestions for Authors

In this manuscript the authors investigated the level of alarmines in CSU patients. Thereby, the authors found increased levels of IL-33 and TSLP. The manuscript is well written an informative.

However, in my opinion the introduction and the discussion is a bit to focused, and thus needs to be revised. The authors just focus their introduction on CSU and just mention that mast cells and alarmins such as IL-33 are somehow involved in the pathogenesis of CSU. However, mast cells and the investigated alarmines such as IL-33 and TSLP are described as essential for the development of several diseases of the skin. Given the central role of these alarmines in this manuscript and to underpin the common importance of IL-33 and TSLP and mast cells in skin inflammation the authors should mention this and therefore should cite appropriated reports such as:

Hoppe et al., DOI: 10.1016/j.jaci.2020.01.036

Drube et al.; DOI: 10.4049/jimmunol.1700556

Hueber et al; DOI: 10.1002/eji.201041360

The same should be done for IL-25 and TSLP

In the discussion the authors don’t give a potential model which explains the role of the investigated alarmines in CSU. For instance: the fact that IL-33 and TSLP are significantly increased in ther serum of CSU patients the authors should also write how IL-33 and TSLP (e.g. involved cells....) could mediate CSU. Since there are several publications about the action of IL-33 and TSLP in other skin diseases the authors should provide a model which could explain the role of IL-33 and TSLP in CSU patients.

To increase the interest of scientist which are not only interested in reading the next study about the serum level of alarmines, the authors should give such informations.

Reviewer 2 Report

Comments and Suggestions for Authors

The authors have evaluated alarmins and chronic urticaria.

The major comment here is the study design. Just identification of urticaria is easy on clinical examination. What is the justification for biomarkers for diagnosis? It is a different issue with prognosis and it would have been useful to make a statement on biomarkers and disease severity and prognosis.

It would be very useful to include a separate group on urticarial vasculitis

The ROC analysis would help to identify the optimal cut-offs for the cytokines. Does the clinical behavior of the patients change in subjects with lower-than-cut-off IL-33 versus higher-than-cut-off IL-33 levels?

A dose-response relationship is important to demonstrate to suggest that these alarmins are important and is not a simple bystander phenomenon. In IL-33 and DLQI, the values increase from moderate to important but decrease thereafter. It is also unclear how the values changed in IL-33 from Figure 16 and Figure 17.  Important has higher levels than v.important. But in Figure 17, the values don't correlate with Figure 16. 

Comments on the Quality of English Language

English language can be tempered and too many adjectives can be avoided. The writing can be kept scientific and not literary

Reviewer 3 Report

Comments and Suggestions for Authors

Very interesting manuscript with the study of biomarkers in urticaria, alarmins. Much has been studied about them in diseases such as atopic dermatitis, although none have gone beyond phase II to demonstrate effectiveness. Urticaria is pathophysiologically close to atopic dermatitis, although perhaps the IL33 that has been demonstrated in research may be of interest. The question is whether etokimab (monoclonal antibody against this cytokine) will work in chronic urticaria; it has already been observed that it will not in atopic dermatitis. There are other diseases such as nasal polyposis that have also considered this drug, although with these results. Alarmins are important, but I doubt that blocking them would be relevant in clinical practice. The study is well designed and reflected in the manuscript, but perhaps the discussion should be expanded in relation to IL33 and publications about it; nothing is mentioned about etokimab either.

Reviewer 4 Report

Comments and Suggestions for Authors

The paper entitled “The Alarmin Triad - IL-25, IL-33, and TSLP - Serum Levels and Their Clinical Implications in Chronic Spontaneous Urticariais informative from the clinical point of view, but there are some concerns that need to be addressed as follows.

Major concerns

1.    If the manuscript exceeds the limit on the number of figures allowed, the authors should submit the less critical data to a supplementary file.

2.    Based on the findings of high serum levels of IL-33 and thymic stromal lymphopoietin (TSLP), the authors should discuss the cellular players of chronic spontaneous urticaria (CSU).

3.    In connection with this, I suggest the authors confirm serum levels of itch-related cytokines other than IL-33 (eg, IL-4, IL-13 and IL-31).

Remarks:

In the pathogenesis of CSU, IL-33 is known to augment IgE-mediated histaminergic itch through mast cell activation and IL-13 production.

On the contrary, TSLP-elicited basophils are known to express IL-31 and contribute to pruritus in CSU through non-IgE-mediated mechanisms.

Round 2

Reviewer 4 Report

Comments and Suggestions for Authors

I think this paper is suitable for publication in IJMS.